# Multi-Modality and Multi-Scale Attention Fusion Network for Land Cover Classification from VHR Remote Sensing Images

**Tao Lei** [1,2,*], **Linze Li** [1,2], **Zhiyong Lv** [3], **Mingzhe Zhu** [4], **Xiaogang Du** [1,2] and **Asoke K. Nandi** [5,6]

1. Shaanxi Joint Laboratory of Artificial Intelligence, Shaanxi University of Science and Technology, Xi'an 710021, China; lilinze@sust.edu.cn (L.L.); duxiaogang@sust.edu.cn (X.D.)
2. School of Electronic Information and Artificial Intelligence, Shaanxi University of Science and Technology, Xi'an 710021, China
3. School of Computer Science and Engineering, Xi'an University of Technology, Xi'an 710048, China; zhiyongLyu@xaut.edu.cn
4. School of Electronic Engineering, Xidian University, Xi'an 710071, China; zhumz@mail.xidian.edu.cn
5. Department of Electronic and Electrical Engineering, Brunel University, London UB8 3PH, UK; asoke.nandi@brunel.ac.uk
6. School of Mechanical Engineering, Xi'an Jiaotong University, Xi'an 710049, China
* Correspondence: leitao@sust.edu.cn

**Abstract:** Land cover classification from very high-resolution (VHR) remote sensing images is a challenging task due to the complexity of geography scenes and the varying shape and size of ground targets. It is difficult to utilize the spectral data directly, or to use traditional multi-scale feature extraction methods, to improve VHR remote sensing image classification results. To address the problem, we proposed a multi-modality and multi-scale attention fusion network for land cover classification from VHR remote sensing images. First, based on the encoding-decoding network, we designed a multi-modality fusion module that can simultaneously fuse more useful features and avoid redundant features. This addresses the problem of low classification accuracy for some objects caused by the weak ability of feature representation from single modality data. Second, a novel multi-scale spatial context enhancement module was introduced to improve feature fusion, which solves the problem of a large-scale variation of objects in remote sensing images, and captures long-range spatial relationships between objects. The proposed network and comparative networks were evaluated on two public datasets—the Vaihingen and the Potsdam datasets. It was observed that the proposed network achieves better classification results, with a mean *F1-score* of 88.6% for the Vaihingen dataset and 92.3% for the Potsdam dataset. Experimental results show that our model is superior to the state-of-the-art network models.

**Keywords:** land cover classification; multi-modality data fusion; deep learning; multi-scale spatial contextual information

## 1. Introduction

Because VHR remote sensing images can provide more details of ground targets, they have been widely used for land cover classification and recognition under complex scenes. During land cover classification, it is a challenge to assign all pixels in remote sensing images to different semantic categories. In contrast to single target recognition, in land cover classification, multiple targets in the image scene can be recognized at the same time, and the spatial distribution of ground targets cam also counted. Therefore, VHR (very high-resolution) remote sensing images have been extensively applied for building recognition [1], road extraction [2], change detection [3], urban engineering [4], etc.

In recent years, with the rapid development of deep learning techniques [5], convolutional neural networks (CNNs) [6] can provide hierarchical feature representation and learn deep semantic features, which are important and useful for improving model performance. Therefore, CNNs have achieved significant success in the field of computer

vision, including target detection [7], image segmentation [8], image classification [9], and visual reconstruction [10]. Benefitting from the rapid development of CNN, researchers have devised with a large number of improved CNN models for image segmentation, such as FCN [11], SegNet [12], U-Net [13], PSPNet [14], DeepLab v1–v3 [15–17], and CPNet [18]. Because land cover classification can be considered to be an image semantic segmentation task, these popular segmentation networks can also be used for land cover classification.

VHR remote sensing images completely capture the ground details and help to accurately analyze the objects in the scene. However, two problems exist in the application of VHR remote sensing images to land cover classification:

- The VHR remote sensing images usually contain large complex scenes and a large number of spectral features can be easily confused; thus, it is arduous for CNN to obtain sufficient features only from spectral data. For example, in Figure 1, segmentation models are often confused by factors such as shadows and occlusion.
- Simple multi-scale extraction modules can no longer adapt to complex VHR remote sensing images, because the scale information of ground targets is quite different.

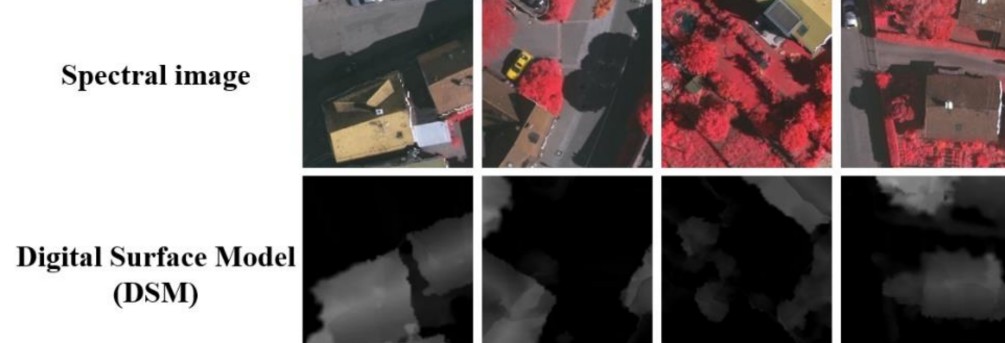

**Figure 1.** The top row is the spectral image of VHR remote sensing images, and the bottom row is the corresponding DSM image. The first two columns are buildings and roads in shadows. The latter two columns are trees and low vegetation with extremely similar spectral characteristics. The digital surface model avoids the interference of shadow, occlusion, and other factors.

It is crucial to improve the accuracy of land cover classification to solve these two problems. In this regard, there is significant potential for the utilization of different modalities' image features. Fortunately, the features of geographic information, such as digital surface models (DSMs), can well avoid this type of interference [19], which is also demonstrated in Figure 1. To address the first problem, we effectively fuse DSM features and spectral features in our proposed model to promote the accuracy of some targets. In addition, spatial relationship modeling provides additional important information, which can be used to simulate the semantic similarity of uncertain local receptive fields [20]. Refs. [21,22] prove the capture spatial dependence can significantly improve the performance of the semantic labeling of VHR images. Thus, it is of great significance to introduce the spatial relationship between pixels to complement multi-scale features. To address the second problem, we capture the long-range dependencies in the learning process to model the spatial relationship between pixels and introduce this spatial relationship into multi-scale feature extraction to enhance the precision of the results.

In this study, we proposed a multi-modality and multi-scale attention fusion network, namely MMAFNet, for land cover classification from VHR remote sensing images. The contributions of this study are briefly summarized as follows:

- We designed a multi-modality fusion module to fuse richer features and avoid redundant features of IRRG and nDSM images. This addresses the problem of low accuracy of land cover classification from VHR remote sensing images caused by the weak feature representation ability of single modality data.

- We present a novel multi-scale spatial context enhancement module that considers the advantages of both ASPP (Atrous Spatial Pyramid Pooling) and a non-local block to improve image feature fusion, which successfully addresses the problem of a large difference in target scale in VHR remote sensing images.

The remainder of the paper is organized as follows. Section 2 describes land cover classification networks in remote sensing images and the recent advances. Section 3 explains our proposed network structure in detail. Section 4 presents the experiments, and a thorough analysis and discussion of the results. Section 5 provides conclusions and future prospects.

## 2. Related Work

In this section, the progress of remote sensing image land cover classification is reviewed. We list some representative work transitioning from traditional methods to deep learning methods. In addition, from the perspectives of multi-modality data fusion and multi-scale feature extraction methods, we review the latest developments in remote sensing image land cover classification in recent years.

### 2.1. Traditional Methods

In earlier studies of land cover classification, the structure of combining a feature extractor and a classifier was commonly used by researchers [23,24]. The feature extractor locally extracts spatial and texture features from the image, and then each pixel is allocated by a classifier. For example, Tarabalka et al. [23] further merged the segmentation map obtained by the watershed segmentation algorithm into the spectral-space classifier to improve the classification accuracy. Fan et al. [24] proposed a single-point iterative weighted classifier based on prior knowledge to guide unsupervised algorithms to obtain the best clustering results. Other traditional methods, such as fuzzy c-means clustering (FCM) [25], support vector machines (SVMs) [26], and neural networks [27], have been applied to remote sensing image analysis. These methods improve the accuracy of the land cover classification to a certain extent. However, a common disadvantage is that their performance depends heavily on feature selection.

### 2.2. Popular Segmentation Networks

Driven by deep learning techniques, pixel-level segmentation has made significant progress in the field of remote sensing images, because it can automatically learn high-level semantic features from images in an end-to-end manner. Long et al. [11] first developed a fully convolutional network (FCN) for image semantic segmentation. The FCN can accept any size of input image, which ensures the robustness of the model. However, the segmentation result of the FCN is not precise due to the loss of target edge information in down-sampling.

To solve this problem, Badrinarayanan et al. [12] proposed the SegNet network using a fully symmetric encoder-decoder structure. SegNet retains the index of the maximum value during down-sampling and uses it to recover the edge position of the target. The U-Net network proposed by Ronneberger et al. [13] also has a similar symmetric structure to that of SegNet. U-Net fuses the corresponding features of the encoding and decoding stages to refine the segmentation accuracy. Because these networks are relatively simple [28] and do not consider the global context information, the segmentation accuracy still needs to be improved.

Global context information refers to the relationship between the pixels and the surrounding pixels, which improves the accuracy of the model in identifying semantic categories. PSPNet [14] uses a pyramid pooling module to aggregate global context information from different regions, but it is computationally inefficient. DeepLab v3+ [17] incorporates the convolutions of various dilation rates and a global average pooling into the network to obtain a multi-scale context feature. However, DeepLab v3+ cannot capture

long-range dependencies in the learning process, and cannot reduce the interference in complex scenes.

### 2.3. Multi-Modality Data Fusion

Compared with natural images, remote sensing images are more complicated. However, DSM features with rich geographic information can reduce the interference caused by complex scenes. At present, the feature fusion of multispectral images and depth images has become a research focus for enhanced feature representation of land cover classification. Sherrah [29] proposed a network for independently extracting DSM features and fine-tuning it on VHR remote sensing images. To further improve the efficiency of fusing DSM information, Cao et al. [30] proposed a lightweight deep separable convolution module to extract the features of depth images, and designed a variety of fusion strategies to explore the best network structure.

However, simply superimposing the spectral and DSM images as the input of the network does not make full use of the relationship between the multi-modality information, and introduces redundant features in the training phase [31]. Therefore, a more reasonable and effective fusion of multi-modality features acts a pivotal step in improving the accuracy of land cover classification. There are two mainstream multimodal feature fusion methods.

In the first, RGB and DSM information is fused before feature extraction, and a variety of attention mechanisms are introduced to establish the multi-modality feature association in the learning stage. For instance, Liu et al. [32] used DenseNet [33] as the backbone network and introduced a spatial and a channel attention module in the process of feature extraction, which not only makes full use of the internal weight features of the image, but also strengthens the information correlation within the image. However, the model obtained by this method has high complexity, so it is difficult for the model to converge during the training phase.

The second approach is an early fusion method in which spectral image features and depth image features interact continuously in the encoding stage. For example, Audebert et al. [34] used two deep networks to extract RGB and DSM image features, and proposed an early fusion network structure. This network structure uses a residual connection that provides feature weighting, and merges the spectral features and the depth features together for the final up-sampling stage. The cost of this early fusion method is high, and it involves numerous parameters in the model. However, due to its better performance, many of the current networks use this parallel feature extraction structure as a reference. Peng et al. [35] used the idea of dense connection to improve the early fusion strategy; however, the fusion strategy focuses on the use of spectral features, resulting in a lower efficiency of features fusion.

### 2.4. Multi-Scale Feature Extraction Methods

In the VHR remote sensing images, the target usually has a large difference in scale, leading to performance degradation for the land cover classification task. To solve this problem, at present multi-scale feature extraction is often achieved by superimposing convolutions with different convolution kernel sizes. The effect of multi-scale feature fusion is reduced due to the simple fusion design. In spite of this, image semantic segmentation with multi-scale information can still improve the results. Moreover, the low segmentation accuracy caused by the large difference in target scales can be upgraded by introducing context information of different scales. Marmanis et al. [36] used pre-trained VGG-16 [37] as the backbone network, and designed a multi-scale pixel-level segmentation architecture fusing a fully convolutional network and transposed convolution. Yu et al. [38] proposed an improved version of PSPNet and applied a pyramid parsing network to extract multi-scale features from VHR remote sensing images. These methods can incorporate multi-scale features, but they are computationally expensive and their accuracy is relatively low for land cover classification.

To further improve the classification accuracy, numerous researchers employed multi-scale context features in networks [39–43]. For instance, Shang et al. [40] proposed an adaptive method to capture multi-scale information. He et al. [42] proposed a multi-scale attention refining module to enhance the representation ability of the feature map extracted by the depth residual network. In addition, Zhao et al. [43] proposed a pyramid attention module, which introduced the attention mechanism into the multi-scale module for adaptive feature refinement. These methods [39–43] addressed the problem of the excessive target scale difference, but did not introduce a long-range dependence when capturing multi-scale information, and only paid attention to the channel relationship.

The multi-branch spatial-channel attention network proposed by Han et al. [22] and the multi-attention network proposed by Li et al. [21] simultaneously consider the spatial and channel relationship of feature maps. These works [21,22] prove that spatial and channel dependence can improve the performance of the semantic labeling of VHR images.

## 3. Proposed Method

In this section, we first introduce the overall network of MMAFNet, and then recommend the architecture of different modules for land cover classification.

### 3.1. Overall Architecture

MMAFNet mainly includes two parts, a multi-modality fusion module (MFM) based on ResNet50 [44] and a multi-scale spatial context enhancement module (MSCEM); the overall architecture is shown in Figure 2. MFM is used to extract and fuse multi-modality data, and fully learn multi-modality features with an improved early fusion method. MSCEM is used to extract multi-scale features and learn global context information by introducing spatial relationships. The final segmentation result is achieved by performing bilinear interpolation on the connected feature maps. In addition, the residual learning strategy (RSC) is also used in our network.

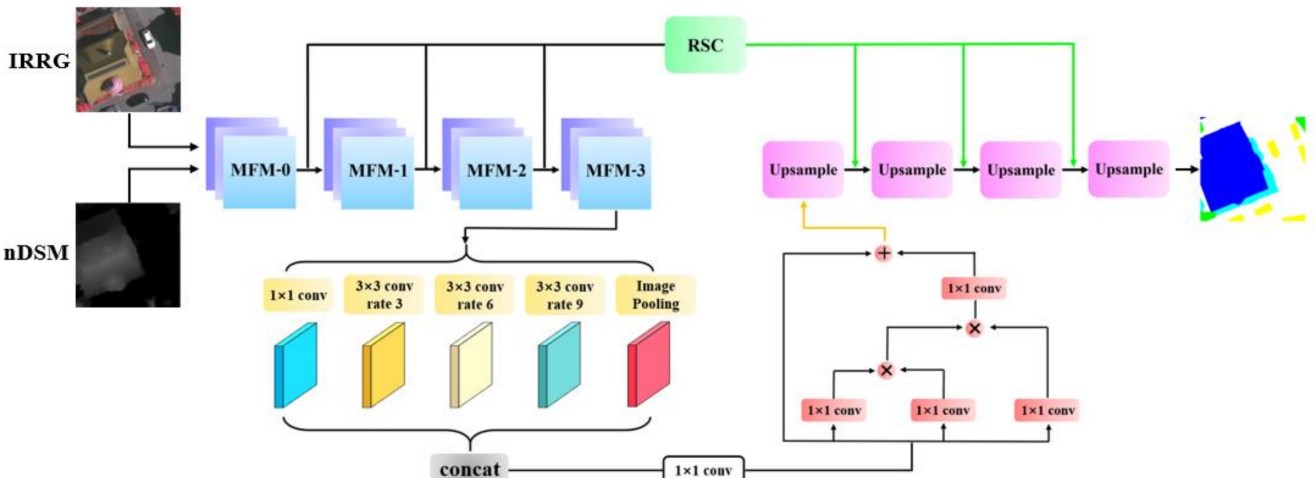

**Figure 2.** The overall architecture of MMAFNet.

### 3.2. Multi-Modality Fusion Module

Large-scale complex scenes and ground details are often included in VHR remote sensing images, and some areas have similar colors and textures. As shown in Figure 1, roads under shadow interference or trees with similar spectral features and low vegetation usually cause the features extracted by CNN to be misclassified. The feature of depth image can help the network reduce the probability of error classification. However, most of the semantic segmentation networks used for RGB-D cannot handle data from different modalities in a balanced manner [45]. When dealing with DSM data, only a simple

encoder is used for its feature extraction. The depth branch only deals with depth-related information, whereas the spectral branch deals with a mixture of depth and spectral data. Although DSM images are single-channel images, DSM information is not fully utilized in this method for complex VHR remote sensing scenes.

To make full use of IRRG and DSM data, we propose a more balanced network structure to process and fuse the information, as shown in Figure 3. MMAFNet uses a pre-trained ResNet50 with image features extracted in the spectral and depth branches, respectively. Each of the aforementioned branches provides a set of feature mappings at each module stage, on top of which we introduce the third fusion branch. The branch is used to process the fused data, and Figure 3a shows that the third branch takes the fusion from the spectral and depth branches as the input before down-sampling.

$$M_0 = I_0 + D_0, \tag{1}$$

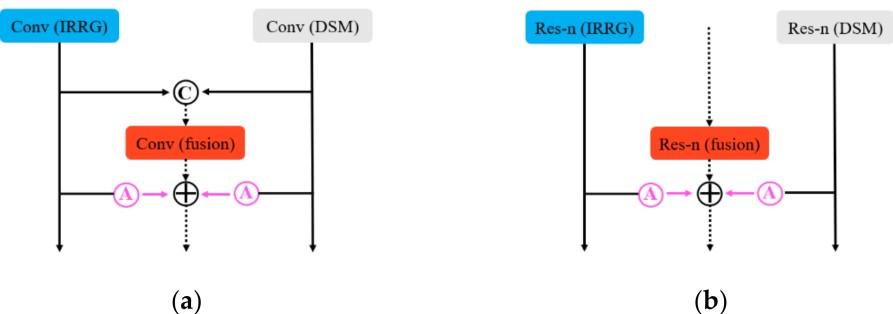

(**a**)             (**b**)

**Figure 3.** Multi-modality fusion module (MFM). (**a**) MFM-0. (**b**) MFM-$n$ ($n \in [1, 3]$).

In the initial stage, we concatenate feature $I_0$ from the IRRG data and feature $D_0$ from the DSM data to obtain the feature map $M_0$ of the third branch, denoted by Equation (1), where $+$ denotes the concatnation.

Residual learning is performed through a convolutional block that utilizes the sum of the feature maps of the other two encoders up to stage MFM-3. Before combining the feature maps, we introduce a module based on channel attention (MCA) that reorganizes the feature maps from both encoders and allocates resources rationally. By obtaining the appropriate weights and adding them together, our network can extract, reorganize, and fuse the information of different modalities to provide useful information. The output detail of MFM-0 is denoted by Equation (2).

$$F_0 = \hat{A}(Conv(I_0)) \oplus \hat{A}(Conv(D_0)) \oplus Conv(M_0), \tag{2}$$

where $\oplus$ denotes pixel-level summation and $\hat{A}$ denotes the module based on channel attention. We introduce this module in detail in the next subsection.

The structure of MFM-$n$ ($n \in [1, 3]$) is shown in Figure 3b, where three ResNets extract features from three branches, and the ability of feature representation is stronger after residual learning. These features are then fused using the same pattern as the MFM-0 phase. Notably, we drop the last down-sampling of the encoding stage. The output details of MFM-$n$ ($n \in [1, 3]$) are denoted by Equation (3).

$$F_n = \hat{A}(Res(I_{n-1})) \oplus \hat{A}(Res(D_{n-1})) \oplus Res(F_{n-1}), \tag{3}$$

This fusion approach fully utilizes the information of different modalities and is more suitable for land cover classification from VHR remote sensing images. The effectiveness of this module has also been proven in ablation experiments.

Module based on channel attention: There is a large quantity of redundant information in the fusion of multi-modality features, and this information is counterproductive to distinguishing targets with high similarity in remote sensing images. In addition, incorrect

classification results of the model occur because the targets are provided with similar distribution patterns but different channel dimensions. In CNN, enhanced channel coding is helpful for image classification tasks [46]. Therefore, to collect features selectively from the spectral and depth branches, we propose a module based on channel attention in the MFM to model the channel relationship, which can be used to enhance the feature discrimination in the channel domain, so that we can further understand complex high-resolution remote sensing scenes.

We regard the input feature map $A = [a_1, a_2, \ldots, a_c]$ as a combination of channels $a_i \in R^{H \times W}$. First, we obtain a vector $G \in R^{1 \times 1 \times C}$ and its $k^{th}$ element after performing global average pooling (GAP).

$$G_k = \frac{1}{H \times W} \sum_i^H \sum_j^W A_k(i, j), \tag{4}$$

The operation integrates the global information into the vector G. Then the vector is transformed to $\hat{G} = O_2(Activation(O_1(G)))$, where $O_1 \in R^{1 \times 1 \times C/2}$ and $O_2 \in R^{1 \times 1 \times C}$. These are two fully connected convolutional layers, and an *Activation* function is added after $O_1$, which creates a channel dependence on feature extraction. Further, $\hat{G}$ is activated by the *Sigmoid* function $\sigma(\cdot)$, which is constrained to $[0, 1]$. Finally, $A$ and $\hat{G}$ are multiplied to obtain $\hat{A}$.

$$\hat{A} = \sigma(o_2(Activation(o_1(GAP(A))))) \otimes A, \tag{5}$$

*ReLU* remaps the original channels into new channels, adding nonlinear elements in an adaptive form and making the network fit better. In the process of network learning, the module suppresses useless features and recalibrates the weights on a more meaningful feature map for further learning.

### 3.3. Multi-Scale Spatial Context Enhancement Module

For the task of land cover classification using VHR remote sensing images, first, the proportions of targets are very different, thus requiring feature maps to capture multi-scale information. Second, each target is geometrically related, which requires a non-local idea [47]. Third, a large receptive field is needed for high-level semantic features. It is not sufficient to stack four $3 \times 3$ convolutional layers. The spatial relationship is introduced to obtain fine-grained context features. Therefore, we combine the advantages of ASPP (Atrous Spatial Pyramid Pooling) and a non-local block, and propose a multi-scale spatial context enhancement module (MSCEM) to replace the simple multi-scale extraction module.

In the last stage of encoding, the multi-scale feature map is captured completely by ASPP. Because we drop the last down-sampling, the atrous convolution of different atrous rates is operated on the same feature map and fused with the output, and the fusion should cover the whole feature maps. Thus, we combine the $3 \times 3$ convolutions with atrous rates 3, 6, and 9, and a regular $1 \times 1$ convolution for multi-scale information extraction. An image average pooling is also attached to integrate global contextual information. Such a combined operation has better efficiency and performance without increasing the number of parameters.

Applying a global contextual remote dependency strategy is crucial in the land cover classification of multi-classified remote sensing images. To thoroughly utilize the spatial information of multi-scale feature maps, we borrowed the idea of non-local block [48] after the integration of multi-scale information. DSM data also provide geographical geometric attributes for some classes in remote sensing images. Spatial relationships can enhance the local properties of feature maps by aggregating dependencies on other pixel locations. For targets with similar semantic features, the strategy of linking contexts clearly enhances the intra-class feature relevance by combining global and local information, and increasing the accuracy of the results of multi-category land cover classification tasks.

As shown in Figure 4, the feature map is fused with multi-scale information through the ASPP module and then downscaled to 256 channels using $1 \times 1$ convolution into the non-local module. The multi-scale spatial context enhancement module not only extracts and integrates multi-scale feature maps, but also improves and enhances intra-class features by using the remote dependency of the spatial context, which effectively distinguishes different semantic classes in remote sensing images.

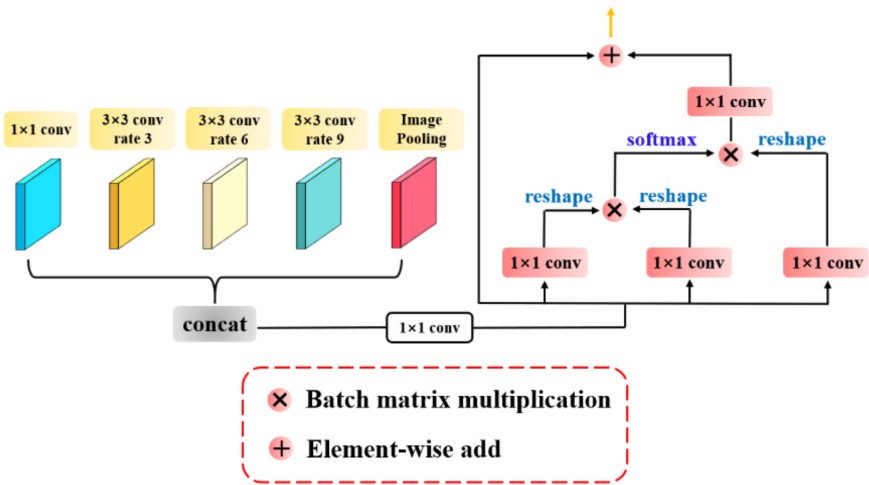

**Figure 4.** Multi-scale spatial context enhancement module.

### 3.4. Residual Skip Connection Strategy

Segmentation networks based on encoder-decoder structures have become increasingly popular in recent years. Because images undergoing constant down-sampling during the feature extraction period lose a large amount of information, it is difficult to provide feature mapping for the decoding stage. A conventional skip connection transfers the low-resolution information at each stage in a duplicate manner to achieve image feature preservation; however, this result generally makes the captured image features blurred. Furthermore, the high-resolution image edge information is not included in the high-level features from learning. To solve these problems, we supplement the skip connection with a post-activation transpose convolution operation. This enhances the overall information of the image in the skip connection and limits the edges of the object to be segmented in the remote sensing image, thus achieving better performance.

As shown in Figure 5, the features of the $(l-1)^{th}$ layer are passed to the $(l+1)^{th}$ layer through a skip connection, and its features are passed via down-sampling to the $l^{th}$ layer and up-sampling to the $(l+1)^{th}$ layer. Thus, the two operations are repeated. This repetition of low-resolution information leads to the blurring of segmentation boundaries. The conventional skip connection passes the high-resolution feature map directly without any convolutional layer learning; thus, the final learned network model cannot effectively map the high-resolution information. In our proposed residual skip connection strategy, we recover the features of the $l^{th}$ layer to the same size as the $(l-1)^{th}$ layer after a transposed convolution learning, and extract the undersampled features of the $(l-1)^{th}$ layer separately from them for summation operations. Finally, this part of the features is learned again using depth separable convolution and transmitted to the $(l+1)^{th}$ layer after up-sampling. This step not only controls the number of parameters but also allows the features to reach a higher level. The detail of the residual skip connection strategy is shown in Equation (6).

$$f_{l+1} = DSC(Activation(Tconv(f_l)) \oplus f_{l-1}), \tag{6}$$

where $f_l$ denotes the feature of the $l^{th}$ layer. *Tconv* denotes the transposed convolution. *Activation* denotes the *ReLU* activation function. *DSC* denotes the depth separable

convolution. $\oplus$ denotes the pixel-level sum operation. $f_{l-1}$ denotes the feature of the $(l-1)^{th}$ layer without down-sampling, and $f_{l+1}$ denotes the result after residual skip connection processing.

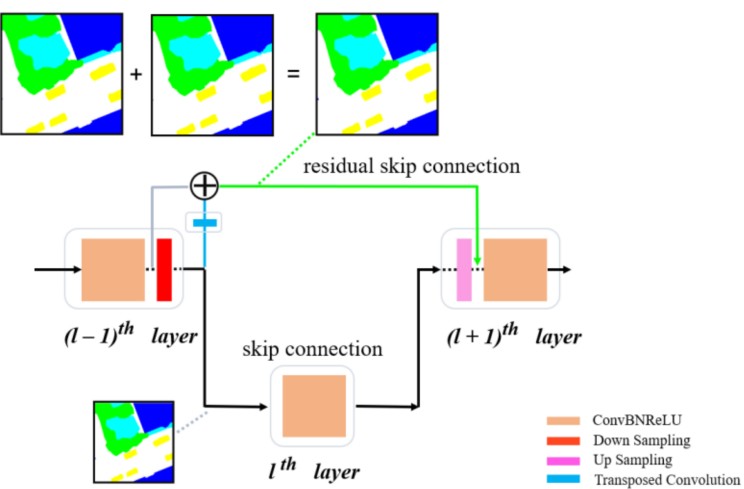

**Figure 5.** The architecture of the residual skip connection.

For small targets of remote sensing images, such as cars, the target feature information is lost with the decrease in resolution after down-sampling. The traditional skip connection is not able to better comprehend the small target information in the small feature map, and the residual skip connection operation passes the small target pool's higher-level global features to the decoder. For large targets, such as buildings, the approach operates in the same manner, improving intra-class information while more effectively limiting edges.

## 4. Experimental Results and Analysis

To verify the effectiveness of MMAFNet, we used the Vaihingen and Potsdam datasets from the ISPRS [49] to carry out land cover classification experiments. We used a quantitative accuracy assessment method to evaluate the classification results based on overall accuracy (OA) and *F1-score*. In addition, we compared the performance of the model with those of remote sensing image land cover classification networks based on deep learning in recent years. Furthermore, we designed ablation experiments to validate the capability of MMAFNet.

### 4.1. Dataset Description

The datasets were provided by Commission III of the International Society for Photogrammetry and Remote Sensing (ISPRS). The VHR true orthophoto (TOP) slices, DSMs, and corresponding ground truth (GT) of two German urban regions are contained in the dataset.

Thirty-eight images in Potsdam dataset [50] have three bands: near-infrared (IR), red (R), and green (G). Moreover, the corresponding normalized DSM (nDSM) is also presented. The image slices have a spatial resolution of 5 cm and are all 6000 × 6000 pixels in size. Impervious surfaces, buildings, low vegetation, trees, cars, and clutter are marked on each pixel in 24 images. Better segmentation performance can be obtained for IRRG images compared to RGB images [51]. Therefore, to further improve the segmentation accuracy, we used both IRRG and nDSM data types. Image serial numbers 5_12, 6_7, and 7_9 were selected for validation; image serial numbers 5_10 and 6_8 were selected for predicting; and the remaining images were selected for training.

Thirty-three images with an average size of 2500 × 2100 pixels and a spatial resolution of 9 cm are included in the Vaihingen dataset [52]. Only 16 images in the dataset have GT, and each image has the same band and label as those of the Potsdam dataset. Considering the accuracy, we also used both IRRG and nDSM data types. We chose five images as the

prediction set to evaluate our network, namely 11, 15, 28, 30, and 34. Three images were used as the validation set, namely 7, 23, and 37, and the remaining images were used for training. Figure 6 shows the sample images from these two datasets.

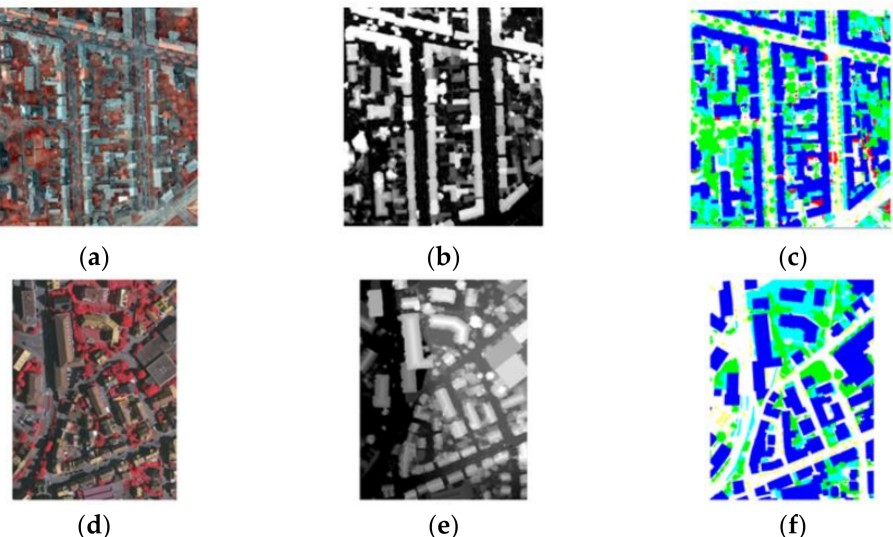

**Figure 6.** Sample images of Potsdam and Vaihingen datasets, digital surface models, and their corresponding labels. (**a**) Potsdam TOP, (**b**) Potsdam DSM, and (**c**) Potsdam GT, (**d**) Vaihingen TOP, (**e**) Vaihingen DSM, and (**f**) Vaihingen GT.

The original images from the dataset were required to be cropped to fit MMAFNet input due to the limitation of GPU memory. Thus we cropped each image to a size of $256 \times 256$ pixels with 128 pixels of overlap, and the final predictions were stitched together. To avoid over fitting, we use random flip and random rotation to augment the data. As a result, the network is able to effectively prevent over fitting during training, and the final model also has strong robustness.

### 4.2. Training Details

MMAFNet was built using the deep learning framework PyTorch. ResNet50 pretrained on ImageNet [53] was used as our backbone network. Our operating system was Windows 10, the processor was an Intel(R) Xeon(R) CPU E51620 v4, and MMAFNet was trained on two NVIDIA GeForce GTX 1080 GPUs with 8 GB of memory each. For network training, the stochastic gradient descent (SGD) optimizer having a momentum of 0.9, weight decay of 0.004, and initial learning rate of $1 \times 10^{-3}$ was used to optimize the network. We used cross entropy as the loss function of the network, set the total epoch and total batch size to 250 and 16, respectively, and multiplied by 0.98 at the end of each epoch.

### 4.3. Metrics

We adopted the OA and *F1-score* as our evaluation metrics to evaluate the performance of the different methods; the major metrics are as follows:

$$F1 = 2 \times \frac{Precision \times Recall}{Precision + Recall}, \tag{7}$$

$$OA = \frac{TP}{TP + TN + FP + FN}, \tag{8}$$

where

$$Precision = \frac{TP}{TP + FP}, \tag{9}$$

$$Recall = \frac{TP}{TP + FN}. \tag{10}$$

where $TP$ is the number of true positives, $TN$ is the number of true negatives, $FP$ is the number of false positives, and $FN$ is the number of false negatives.

### 4.4. Results and Analysis

In this section, we compare MMAFNet with popular networks used for remote sensing images, such as DeepLab v3+ [17], DSMFNet [30], REMSNet [32], DP-DCN [35], and MANet [40]. These networks can be divided into two types: those that introduce the spatial relationships and those that do not introduce spatial relationships. Among these, REMSNet introduces spatial relationships, and none of the other networks introduce spatial relationships. Note that DeepLab v3+ does not use DSM images in terms of data usage.

#### 4.4.1. Results on the Potsdam Dataset

In the experiments on the Potsdam dataset, we calculated the *F1-score*, mean *F1-score*, and OA for each class. The results are indicated in Table 1, and the best results are shown in bold.

**Table 1.** Experimental results on the Potsdam dataset (%). Best results are in bold.

| Method | Imp. Surf. | Building | Low Veg. | Tree | Car | Mean *F1* | OA |
|---|---|---|---|---|---|---|---|
| DeepLab v3+ [17] | 89.88 | 93.78 | 83.23 | 81.66 | 93.50 | 88.41 | 87.72 |
| MANet [40] | 91.33 | 95.91 | 85.88 | 87.01 | 91.46 | 90.32 | 89.19 |
| DSMFNet [30] | 93.03 | 95.75 | 86.33 | 86.46 | 94.88 | 91.29 | 90.36 |
| DP-DCN [35] | 92.53 | 95.36 | 87.21 | 86.32 | 95.42 | 91.37 | 90.45 |
| REMSNet [32] | 93.48 | 96.17 | 87.52 | 87.97 | 95.03 | 92.03 | 90.79 |
| **MMAFNet** | **93.61** | **96.26** | **87.87** | **88.65** | **95.32** | **92.34** | **91.04** |

We first evaluated the performance of MMAFNet on the Potsdam dataset. Table 1 shows the classification results. Our proposed network achieved a mean *F1-score* and OA of 92.34% and 91.04%, respectively, thus outperforming other methods in all evaluation metrics. The Potsdam dataset scenes are relatively complex, with trees and low vegetation being more difficult to classify. Compared to DeepLab v3+, our method improved the classification of trees by 7.0% and achieved a considerable improvement in the classification of other categories. This also verifies that our MMAFNet can capture targets well at different scales using global contextual spatial information, illustrating its feasibility for land cover classification of VHR remote sensing images.

Figure 7 shows a comparison of the segmentation results of MMAFNet and other comparative methods on the Potsdam dataset. The unlabeled original images show that trees and low vegetation are so similar that we cannot use the human eyes to correctly classify them in specific areas. DeepLab v3+, MANet, and DSMFNet are also not accurately recognized. The primary reason for misclassification is that these networks do not introduce spatial relationships to model long-range dependency. However, from the comparison shown in the red dashed box, MMAFNet achieves the best segmentation results on the classification of trees and low vegetation, which also validates the superior performance of our network. In addition, for small targets such as cars, our segmentation performance is also more refined than that of comparative methods, due to the residual skip connection strategy that preserves the information of these small targets. Similarly, for large targets, the correlation of intra-class attributes is enhanced to reduce misclassification.

Figure 8 shows the overall classification results of area 5_10 in the Potsdam dataset. The regions and distribution patterns of different categories can be clearly identified. When applied to confusing pixels, MMAFNet can result in higher accuracy. In particular, in the classification of trees and low vegetation and the recognition of cars, our method achieves an exceptional performance.

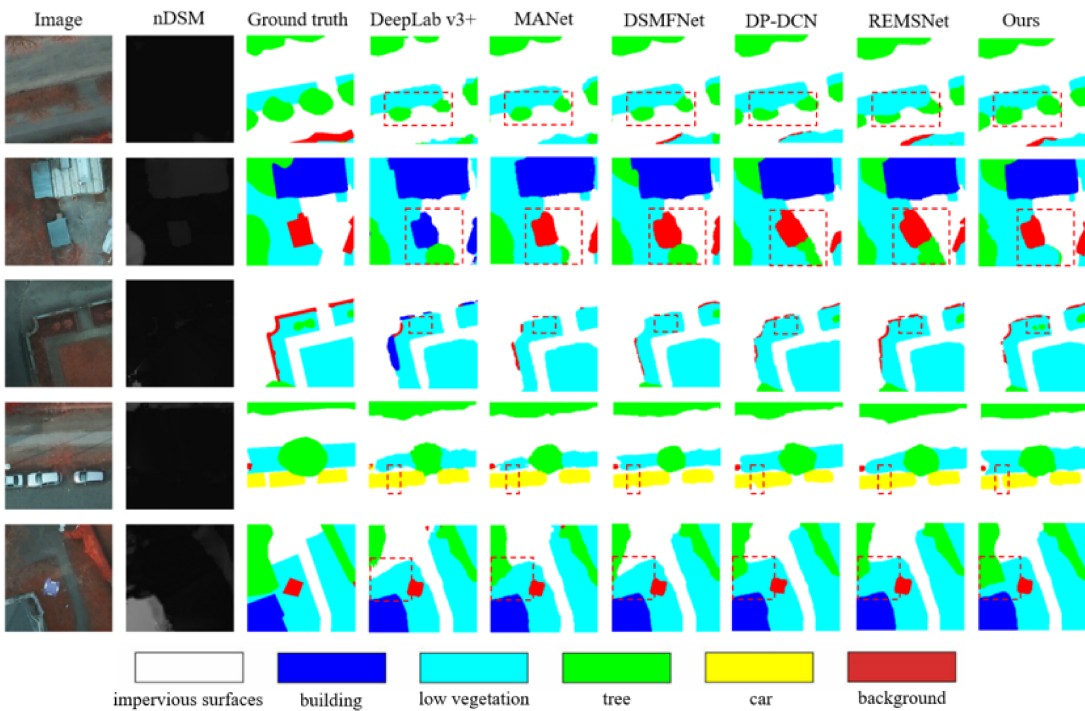

**Figure 7.** Comparison of experimental results for five images in the Potsdam dataset.

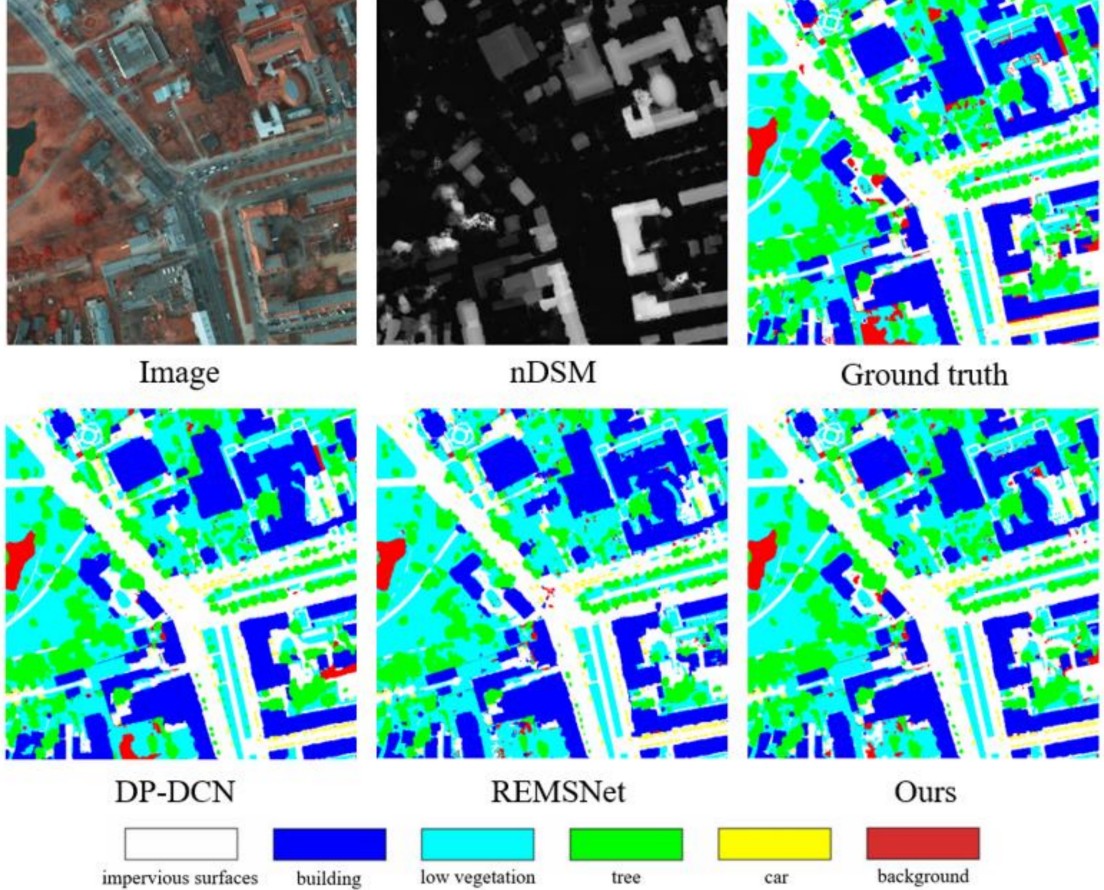

**Figure 8.** The results of MMAFNet and other comparative methods on the Potsdam dataset.

### 4.4.2. Results on the Vaihingen Dataset

In the experiments on the Vaihingen dataset, we calculated the *F1-score*, mean *F1-score*, and OA for each class. The results are represented in Table 2, and the best results are shown in bold.

**Table 2.** Experimental results on the Vaihingen dataset (%). Best results are in bold.

| Method | Imp. Surf. | Building | Low Veg. | Tree | Car | Mean *F1* | OA |
|---|---|---|---|---|---|---|---|
| DeepLab v3+ [17] | 87.67 | 93.95 | 79.17 | 86.26 | 80.34 | 85.48 | 87.22 |
| MANet [40] | 90.12 | 94.08 | 81.01 | 87.21 | 81.16 | 86.72 | 88.17 |
| DP-DCN [35] | 91.47 | 94.55 | 80.13 | 88.02 | 80.25 | 86.89 | 89.32 |
| DSMFNet [30] | 91.47 | 95.08 | 82.11 | 88.61 | 81.01 | 87.66 | 89.80 |
| REMSNet [32] | 92.01 | 95.67 | 82.35 | 89.73 | 81.26 | 88.20 | 90.08 |
| **MMAFNet** | **92.06** | **96.12** | **82.71** | **90.01** | **82.13** | **88.61** | **90.27** |

From the experimental results, it can be seen that the results of MMAFNet are better than those of other methods. The mean *F1-score* and OA were 88.61% and 90.27%, respectively, in the Vaihingen dataset. In particular, for cars, our proposed residual skip connection strategy effectively preserves the information about small objects. We integrate DSM data into the network, which improves the classification of other categories with auxiliary physical spatial height information, and validates that data fusion between different modalities can help in land cover classification. The results show that our network has a stronger capability in complex remote sensing scenes. The multi-scale spatial context enhancement module effectively extracts and identifies features of targets at different scales. Even when the targets occupy a small percentage of a region, and have strong similarity to the surrounding labels, our network can still precisely segment them.

Figure 9 displays the land cover classification outcome of the whole image of the Vaihingen prediction set using different methods. Although trees and low vegetation make classification difficult because of their high similarity, the bounding box with red dashed lines indicates that our proposed network can not only better distinguish high similarity regions, but also retain all the information of small objects. For some factors such as lighting and shadows, MMAFNet can also reduce disturbances to some extent. For example, in the fourth row, trees occluded by shadow can also be correctly classified.

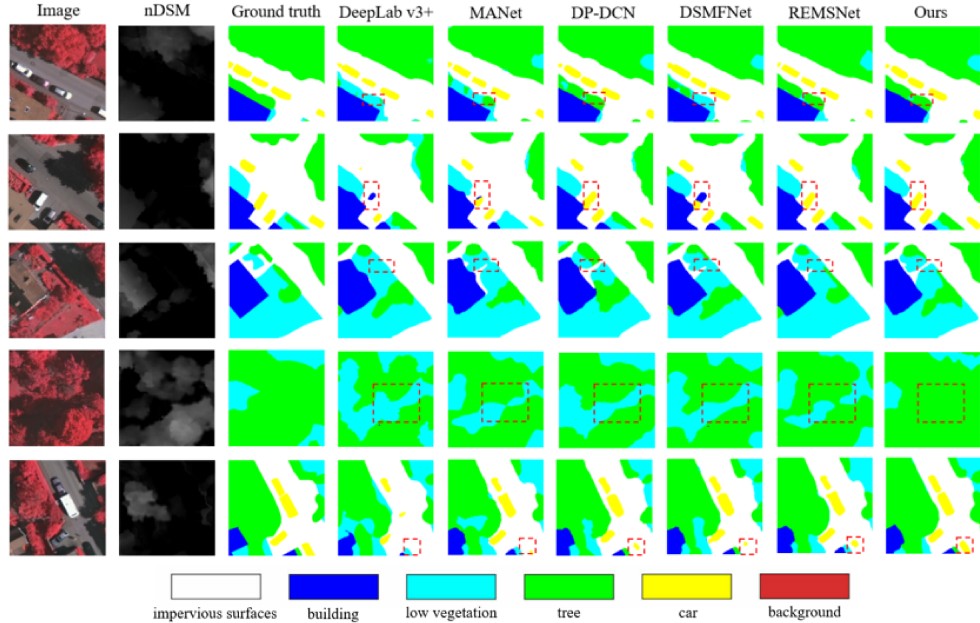

**Figure 9.** Experimental results of MMAFNet and other comparative methods for five images in the Vaihingen dataset.

Figure 10 shows the complete map of the region after predicting the small maps and patching them together. From the final prediction maps, the segmentation maps of the experimental results for different models are very similar. However, it can be concluded from the experimental evaluation metrics that our method has better segmentation performance.

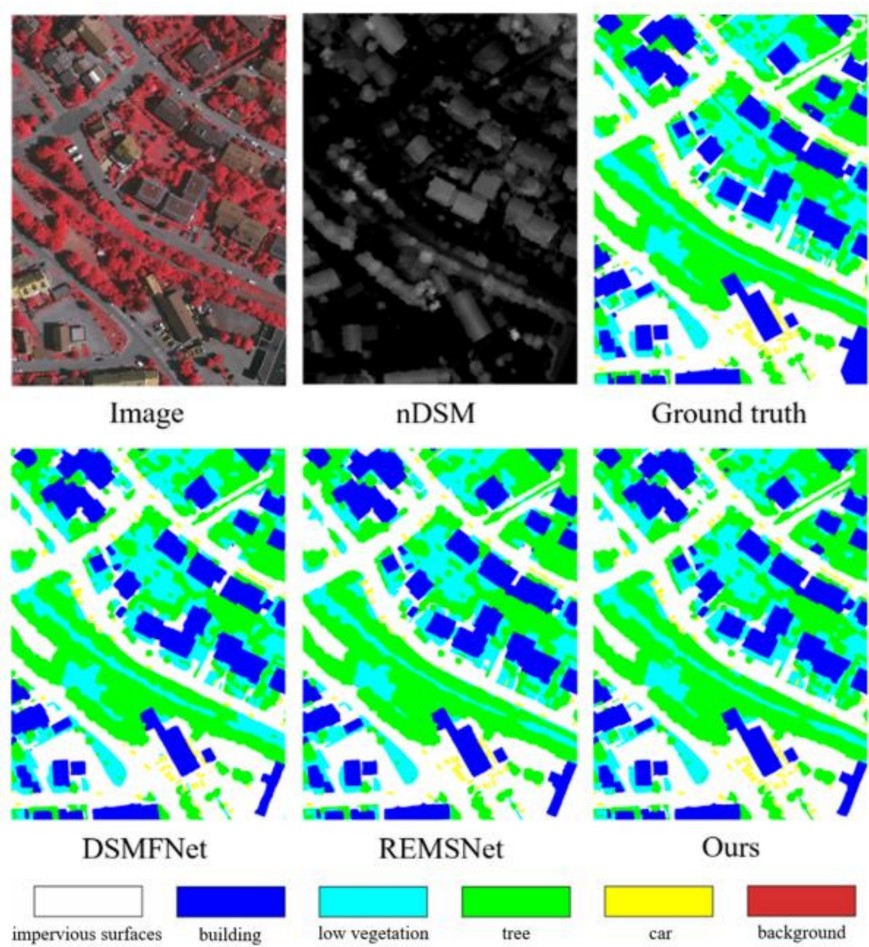

**Figure 10.** The results of MMAFNet on the Vaihingen dataset are shown and compared.

### 4.4.3. Ablation Experiment

We decomposed and combined the proposed network modules and further validated the effectiveness of the different modules using *F1-scores* and OA. The ablation experiments were carried out using the Vaihingen dataset.

In Table 3, we present four comparative models, including Res50, Res50+MFM, Res50+MSCEM, and Res50+RSC, to demonstrate the advantages of the proposed MMAFNet.

**Table 3.** Quantitative analysis of a single module ablation experiment on the Vaihingen dataset; the best result (%) is shown in bold.

| Models | Imp. Surf. | Building | Low Veg. | Tree | Car | Mean *F1* | OA |
|---|---|---|---|---|---|---|---|
| Res50 | 86.94 | 89.67 | 75.83 | 84.42 | 77.40 | 82.85 | 84.98 |
| Res50+MFM | 88.15 | 93.84 | 76.49 | 86.48 | 78.02 | 84.60 | 86.66 |
| Res50+MSCEM | 88.79 | 93.09 | 79.79 | 85.55 | 80.38 | 85.52 | 87.35 |
| Res50+RSC | 90.11 | 92.97 | 80.24 | 86.04 | 81.14 | 86.10 | 87.82 |
| **Res50+MFM+MSCEM+RSC(MMAFNet)** | **92.06** | **96.12** | **82.71** | **90.01** | **82.13** | **88.61** | **90.27** |

The Res50 means a baseline that includes two pre-trained ResNet50, which extracts features from different modalities and fuses these features after the last res block. The

result comes from continuous up-sampling. In the feature extraction process, we did not perform any interaction between different modalities. We chose this model as the baseline for ablation experiments.

Res50 + MFM is a baseline that includes the multi-modality fusion module, which verifies the effectiveness of the multi-modality fusion module. In the process of feature extraction, the two ResNet50s use the attention mechanism to allocate feature resources and continuously carry out information fusion. The third ResNet50 is introduced to handle the fusion branch. In the decoding stage, continuous up-sampling is performed to obtain the final segmentation map.

Res50 + MSCEM is a baseline that includes the multi-scale spatial context enhancement module, which verifies the effectiveness of the multi-scale spatial context enhancement module. In the encoding stage, the feature map obtained by the baseline is the input to the multi-scale spatial context enhancement module and extracted the spatially enhanced multi-scale information. The decoding stage is continuous up-sampling until we get the output.

Res50 + RSC verifies the effectiveness of the residual skip connection strategy in encoder-decoder feature fusion and information retention. Res50 uses the residual skip connection strategy in up-sampling to get the the segmentation result.

Finally, MMAFNet integrated all the modules together, named Res50 + MFM + MSCEM + RSC. All the results of the ablation experiment are shown in Table 3 and Figure 11.

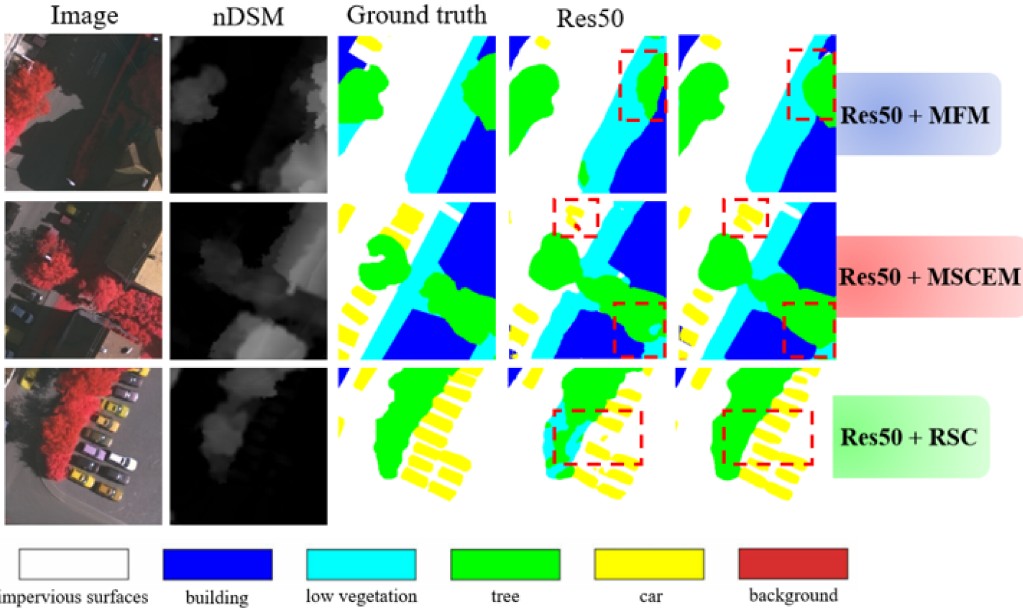

**Figure 11.** Comparison of ablation experiment results of different modules. The first row is a visual comparison between the baseline and the result of adding the multi-modality fusion module. The second row shows that the results verify the benefits of the multi-scale spatial context module. The third row shows the visualization results of the baseline using the residual skip connection strategy.

The results in Table 3 show that the mean *F1-score* of Res50+MFM is 1.8% higher than that of Res50, and the OA is 1.7% higher. The introduction of the multi-modality fusion module proves that making full use of DSM features can significantly improve the classification accuracy. As shown in the first row in Figure 11, the segmentation result obtained by Res50+MFM is closer to the ground truth.

The mean *F1-score* and OA of Res50+MSCEM are 2.7% and 2.4% higher, respectively, than those of Res50. The multi-scale spatial context enhancement module promotes the performance of the baseline network, effectively captures the multi-scale information in the

image, and reinforces the correlation between different classes. In addition, compared with the second row in Figure 11, the complete classification of car proves that integrating long-range dependencies into multi-scale features can reduce the interference from occlusion and shadow.

The mean *F1-score* of Res50+RSC is 3.3% higher than that of Res50, and the OA is 2.8% higher. Compared with the Res50 in Figure 11, the novel residual skip connection strategy can preserve all features. The comparison of accuracy and vision demonstrates that the residual skip connection strategy can successfully improve the classification results.

In addition, the results of the integration of all modules, i.e., MMAFNet, are 5.8% and 5.3% higher than the initial network model, based on the mean *F1-score* and OA, respectively, verifying that MMAFNet can significantly promote the land cover classification performance of VHR remote sensing images.

## 5. Conclusions

In this study, we devised a multi-modality and multi-scale attention fusion network (MMAFNet) acting on the land cover classification of remote sensing images. MMAFNet uses an encoder-decoder structure with ResNet50 as the backbone network. Three parallel branches extract multi-modality data features, and a module based on channel attention is introduced to integrate image information adaptively before each fusion. While adaptively allocating resources, it reduces the redundant information in the image. The multi-scale spatial context enhancement module is supplemented in the final stage of encoding to extract and augment feature information, thus solving the problem of varying scales of target objects in the land cover classification of remote sensing images. During the feature fusion period between the encoder and the decoder, we use a residual skip connection strategy, with enhanced intra-class attributes and restricted edge contours for larger targets. For smaller targets, we retain all their information. The segmentation performance of MMAFNet is more competitive because of the integration of these modules.

In addition, we validated the performance of MMAFNet on the Vaihingen and Potsdam datasets. The experiments showed that our method outperforms other methods, and the ablation experiments further validated the capability of our proposed different modules. In future work, we will strive to decrease the number of parameters of the model and optimize the model for promoting the segmentation performance. We will also aim to promote the practical application of deep learning in the land cover classification of VHR remote sensing images.

**Author Contributions:** T.L. and L.L. provided the original idea for the study; Z.L. and M.Z. conceived and designed the experiments; L.L. and X.D. performed the experiments and presented experimental analysis; Z.L. and L.L. drew all figures; T.L. wrote the paper and A.K.N. revised the paper. All authors have read and agreed to the published version of the manuscript.

**Funding:** This work was supported in part by Natural Science Basic Research Program of Shaanxi under Grant 2021JC-47, in part by the National Natural Science Foundation of China under Grant 61871259, Grant 61861024, Grant 62031021, in part by Key Research and Development Program of Shaanxi (NO. 2021ZDLGY08-07), National Natural Science Foundation of China-Royal Society: Grant 61811530325 (IEC\NSFC\170396,Royal Society, U.K.), and Natural Science Foundation of Gansu Province of China (No. 20JR5RA404).

**Institutional Review Board Statement:** Not applicable.

**Informed Consent Statement:** Not applicable.

**Data Availability Statement:** The datasets used in this study have been published, and their addresses are https://www2.isprs.org/commissions/comm2/wg4/benchmark/2d-sem-label-potsdam/ (accessed on 30 January 2021) and https://www2.isprs.org/commissions/comm2/wg4/benchmark/2d-sem-label-vaihingen/ (accessed on 30 January 2021).

**Acknowledgments:** The author would like to thank the International Society for Photogrammetry and Remote Sensing (ISPRS) for providing data and all colleagues for their contribution to the land cover results of VHR remote sensing images.

**Conflicts of Interest:** The authors declare no conflict of interest.

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
