# Peer review of "Multi-Modality and Multi-Scale Attention Fusion Network for Land Cover Classification from VHR Remote Sensing Images"

_remotesensing, doi:10.3390/rs13183771_

Round 1
Reviewer 1 Report
Manuscript ID: remotesensing-1307910
Manuscript Title: Multi-modality and Multi-scale Attention Fusion Network for Land Cover Classification from VHR Remote Sensing Images
Idea: A novel multi-modality and multi-scale attention fusion network is designed for land cover classification task in this paper. The research in this field is hot topic and there are many papers appeared in the literature recently.
Comments:
- The English of the paper needs to be improved. There are several grammatical mistakes in the paper making the manuscript less readable and understandable.
- How did the Authors fine-tune the parameters of the designed network? Are the number of layers, the filter size in each layer optimized?
- In Table 2, the Overall Accuracy of REMSNet is very close to the proposed method. How this small change can be translated visually for this dataset? Can this small change make a big distinction in terms of visual inspection?
- Figures are very small, please update the figure sizes in revised manuscript. It is difficult to see the differences in classification results, e.g. Figure 11.
- In Figure 2, the input data on the left side of the image must be labeled.
- There are many deep learning-based classification methods developed in the literature recently. Please, if feasible, replace method [36] (belongs to 2018) with a newly developed method.
Decision: Major Revision
Reviewer 2 Report
The paper is enough well written but it deserves some action related to the English grammar and presentation.
The work mentions the use of the attention mechanism but what is practically done (Equation 5) is the use of a gating mechanism that is different from attention. Generally, the attention mechanisms supply distribution scores that model (some kinds) of probability distribution on some features. Here, the authors have used the sigmoid function with the aim to weight (independently) the different components. The authors need to deal with this point and reformulate the text (and the title) according to this point.
The ablation study is not well presented. In particular, the authors did not well present which kind of ablations are made w.r.t. the proposed model. I think that a table that summarises the ablation will provide useful information to a general readers. In addition, also the table associated to the results should be made more clear.
Generally, the contributions are well presented but the piece of work is mainly an additional (DL) architecture that has limited gains w.r.t. other approaches. Since the multi-source approach is managed by early fusion, the novelty is quite limited. Please, can the authors discuss this point related to recent works like [1,2,3,4,5]
[1] https://ieeexplore.ieee.org/document/9487010/
[2] https://ieeexplore.ieee.org/document/9165842/
[3] https://ieeexplore.ieee.org/document/9340317/
[4] https://ieeexplore.ieee.org/document/9513251/
[5] https://ieeexplore.ieee.org/document/9340261/
Round 2
Reviewer 1 Report
No more comments.